# Translating digital healthcare to enhance clinical management: a protocol for an observational study using a digital health technology system to monitor medication adherence and its effect on mobility in people with Parkinson's

Emma Packer ![ORCID],[1] Héloïse Debelle,[1] Harry G B Bailey,[1] Fabio Ciravegna,[2] Neil Ireson,[3] Jordi Evers,[4] Martijn Niessen ![ORCID],[4] Jian Qing Shi,[5,6] Alison J Yarnall,[1,7] Lynn Rochester,[1,7] Lisa Alcock,[1,8] Silvia Del Din[1,8]

For numbered affiliations see end of article.

**Correspondence to**
Dr Silvia Del Din;
silvia.del-din@newcastle.ac.uk

## ABSTRACT

**Introduction** In people with Parkinson's (PwP) impaired mobility is associated with an increased falls risk. To improve mobility, dopaminergic medication is typically prescribed, but complex medication regimens result in suboptimal adherence. Exploring medication adherence and its impact on mobility in PwP will provide essential insights to optimise medication regimens and improve mobility. However, this is typically assessed in controlled environments, during one-off clinical assessments. Digital health technology (DHT) presents a means to overcome this, by continuously and remotely monitoring mobility and medication adherence. This study aims to use a novel DHT system (DHTS) (comprising of a smartphone, smartwatch and inertial measurement unit (IMU)) to assess self-reported medication adherence, and its impact on digital mobility outcomes (DMOs) in PwP.

**Methods and analysis** This single-centre, UK-based study, will recruit 55 participants with Parkinson's. Participants will complete a range of clinical, and physical assessments. Participants will interact with a DHTS over 7 days, to assess self-reported medication adherence, and monitor mobility and contextual factors in the real world. Participants will complete a motor complications diary (ON-OFF-Dyskinesia) throughout the monitoring period and, at the end, a questionnaire and series of open-text questions to evaluate DHTS usability. Feasibility of the DHTS and the motor complications diary will be assessed. Validated algorithms will quantify DMOs from IMU walking activity. Time series modelling and deep learning techniques will model and predict DMO response to medication and effects of contextual factors. This study will provide essential insights into medication adherence and its effect on real-world mobility in PwP, providing insights to optimise medication regimens.

**Ethics and dissemination** Ethical approval was granted by London—142 Westminster Research

## STRENGTHS AND LIMITATIONS OF THIS STUDY

⇒ This protocol has been developed by a multidisciplinary team of experienced clinicians, researchers and data analysts.
⇒ This study will monitor mobility and assess self-reported medication adherence in the real world, enhancing the ecological validity of results.
⇒ If efficacious, this protocol may be implemented in clinical settings to optimise medication regimens and enhance the quality of life for people with Parkinson's disease.
⇒ Participants will only be required to attend a single clinic-based assessment, with the remainder of the study conducted in their home or community.
⇒ This study will include participants with mild to moderate disease severity, therefore, our conclusions may not reflect the experience of those with more advanced Parkinson's disease.

Ethics Committee (REC: 21/PR/0469), protocol V.2.4. Results will be published in peer-reviewed journals. All participants will provide written, informed consent.
**Trial registration number** ISRCTN13156149.

## INTRODUCTION

Mobility, defined as the ability to freely move about one's environment, is a key contributor to overall health and maintaining independence.[1 2] People with Parkinson's (PwP) experience a loss of mobility due to the cardinal motor symptoms of bradykinesia, rigidity, tremor and postural instability.[3] This loss of mobility is associated with an increased risk of falls and reduced independence; therefore, mitigating mobility loss in PwP is of primary importance.[4]

In PwP, dopaminergic medications (eg, levodopa) are commonly prescribed to alleviate motor symptoms and improve mobility.[3] Medication is often prescribed in multiple doses throughout the day, with adherence to prescribed dosage and timing essential to alleviate motor symptoms and avoid fluctuations, especially as the disease progresses. Complex medication regimens result in suboptimal adherence, poor medication responses and reduced quality of life.[5–7] Modification to regimens often follows patient-reported outcomes and short, infrequent clinical appointments, with clinicians observing patients during specific stages in their regimen (eg, ON/OFF periods). Consequently, clinicians often lack adequate insight into patient's medication habits and daily fluctuations in motor symptoms to appropriately adapt regimens. Understanding medication adherence in PwP and its effect on their everyday mobility and motor symptoms would ensure more effective and personalised patient management, allowing clinicians to optimise dose and frequency of dopaminergic medication.

At present, research exploring medication and mobility in PwP is typically conducted in controlled lab-based or clinical environments, in specific ON/OFF medication states.[8–16] Although some studies have been performed in the real world (ie, daily living, outside clinical or laboratory settings),[17 18] a more granular picture is required.

Digital health technology (DHT) presents the tools to explore this, by allowing for the continuous assessment of medication adherence and mobility (digital mobility outcomes, DMOs) in the real world.[5 19–22] A variety of DMOs can be used to assess mobility; and of primary interest is walking speed.[23] Walking speed reflects overall health in PwP and declines overtime, with slow walking speed associated with adverse clinical events such as an increased risk of falls;[4] therefore, assessing this DMO can provide insights into medication related changes in overall mobility and health. The COVID-19 pandemic has also highlighted the fragility of clinic-based care for PwP and reinforced the necessity to use DHT to improve current practice.[24]

There are a variety of DHTs which can monitor individuals during their day-to-day lives. For example, the Personal KineticGraph (Global kinetics Corp, Australia), a wrist-worn device, can send medication reminders and continuously monitor motor symptoms such as bradykinesia, tremor and dyskinesia.[25 26] However, technology such as this cannot monitor the specific medication ingested or quantify gait characteristics. Future research should, therefore, focus on developing a DHT system (DHTS), which can comprehensively and remotely assess medication adherence and monitor mobility. Data collected using this system would allow for the creation

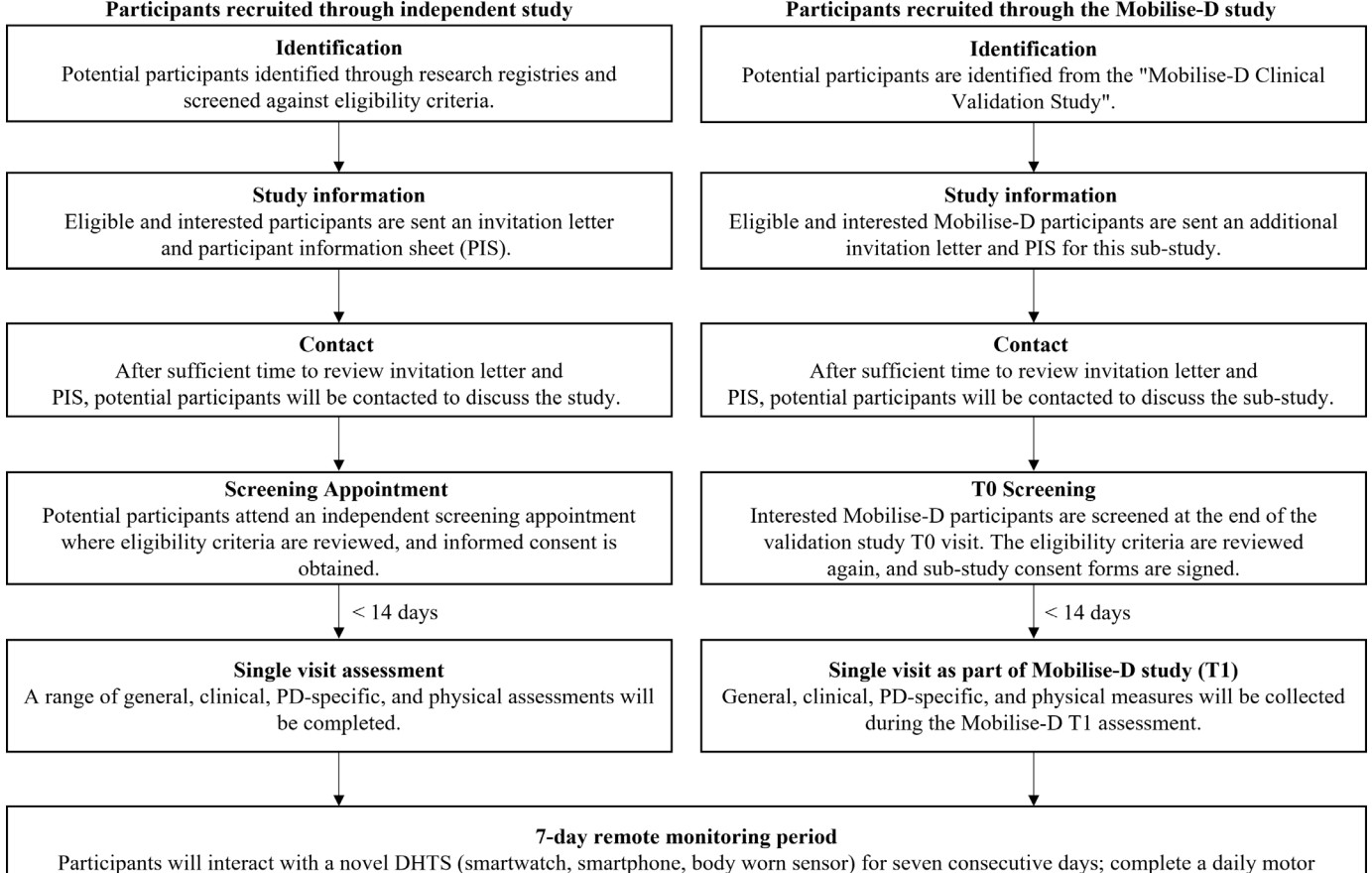

**Figure 1** Flow chart of recruitment and screening procedure. DHTS, digital health technology system; PD, Parkinson's disease.

**Table 1** Inclusion and exclusion criteria

| Inclusion criteria | Exclusion criteria |
|---|---|
| Adults aged 18 or over | Occurrence of any of the following within 3 months prior to informed consent: myocardial infarction, hospitalisation for unstable angina, stroke, coronary artery bypass graft, percutaneous coronary intervention, implantation of a cardiac resynchronisation therapy device, active treatment for cancer or other malignant disease, uncontrolled congestive heart disease (New York Heart Association (NYHA) class >3), acute psychosis, major psychiatric disorders or continued substance abuse |
| Ability to consent and comply with study specific procedures | History consistent with Dementia with Lewy bodies, atypical parkinsonian syndromes (including multiple system atrophy or progressive supranuclear palsy, diagnosed according to accepted criteria) |
| Able to read and write in English | Repeated strokes or stepwise progression of symptoms, leading to a diagnosis of 'vascular parkinsonism' |
| Patients with the clinical diagnosis of Parkinson's according to the Movement Disorder Society criteria[47] | Drug-induced Parkinsonism |
| Hoehn and Yahr stages I–III[48] | |
| On stable Parkinson's medication doses (taking the same medication for 4 weeks or more) | |
| Able to walk 4 metres independently with or without walking aids | |
| Willingness to wear and interact with a smartwatch and IMU, and carry a smartphone | |
| IMU, inertial measurement unit. | |

of data driven models to investigate variations in mobility following medication intake, thus providing greater insights for optimised medication regimens in PwP.

This study aims to collect real-world data using a novel DHTS (smartphone, smartwatch and inertial measurement unit, IMU) to assess self-reported medication adherence, quantify DMOs and model and predict their response to medication. Within this study, specific objectives include: (1) clinically characterise our participants with Parkinson's (eg, Hoehn and Yahr stage, disease duration); (2) collect real-world data with a connected DHTS in PwP; (3) assess real-world, self-reported medication adherence in PwP by recording medication intakes (smartwatch reminders, time of logged intakes and acknowledged medication intake events); (4) quantify real-world DMOs (eg, walking speed) and (5) model and predict response of DMOs to Parkinson's medication.

## METHODS AND ANALYSIS

The Standard Protocol Items: Recommendations for Interventional Trials (SPIRIT) reporting guidelines have been followed within this manuscript[27] (online supplemental material 1). This protocol represents protocol V2.4, April 2022.

## Study design and setting

This cross-sectional, observational study will recruit 55 participants with Parkinson's disease (IRAS Number: 295771). It will be conducted at the Clinical Ageing Research Unit, Campus for Ageing and Vitality, Newcastle University, Newcastle upon Tyne, NE4 5PL, with a remote home and community-based component. The study is cofunded by the Medical Research Council (MRC) Newcastle University, the UK Research Innovation, Engineering and Physical Sciences Research Council (UKRI ESPRC) and the National Institute of Health Research (NIHR). This study is a substudy of the 'Mobilise-D—Clinical Validation Study' (IRAS Number: 289 543 and REC reference: 20/PR/0792), a longitudinal study, which aims to validate a novel digital method for remote monitoring of mobility.[28]

## Study procedure and flow
### Recruitment and screening

In this single-centre study, participants will be identified and recruited between June 2021 and June 2024 through the 'Mobilise-D—Clinical Validation Study' Newcastle upon Tyne site, and independently through local movement disorder clinics and research registries. As recruitment commenced in the height of the COVID-19 pandemic—that is, when individuals were less mobile, less likely to partake in research, and there was insecurity surrounding future lockdowns—an extended recruitment period was afforded to this study. This was to ensure our sample size was achieved and results were representative. Interested and potentially eligible participants will attend a screening appointment during which informed consent will be obtained, and eligibility confirmed. Informed consent will be taken by the chief investigator or a delegated member of the research team. At the beginning of the screening appointment and prior to further discussion, the individual obtaining consent

## Box 1 Outcome measures

**General measures**
⇒ Descriptive measures (year of birth, sex, living arrangement, employment status, years in education, native language, marital status).
⇒ Anthropometric measures (height, mass, shoe size and leg length).
⇒ Health status (current medication and use of mobility aids (eg, walking frame)).

**Clinical measures**
⇒ Disease duration.
⇒ Late-life functional disability instrument.[49]
⇒ Levodopa equivalent daily dose.[50 51]
⇒ Frailty phenotype.[52]

**Parkinson's disease-specific measures**
⇒ Movement Disorder Society-Unified Parkinson's Disease Rating Scale (30).
⇒ Hoehn and Yahr Stages.[48]
⇒ New Freezing of Gait Questionnaire.[53]
⇒ Montreal Cognitive Assessment.[54]

**Physical measures**
⇒ Short Physical Performance Battery.[55]
⇒ Seven-day remote monitoring period (interact with digital health technology system).

**Diary, questionnaire and qualitative questions**
⇒ Motor complications diary (ON-OFF-dyskinesia).
⇒ Usability questionnaire.[38]
⇒ Open-text questions.

will confirm that the potential participant has read the participant information sheet (PIS), and will answer any questions. On agreement to participate in the study, the participant will sign and date the informed consent form and the researcher will witness this, and sign and date the same form (online supplemental materials 2 and 3). The process will then be recorded in the participants' medical records, with the original filed on site and a copy given to the participant. Those recruited through the Mobilise-D study will attend a screening appointment within a Mobilise-D visit and those recruited independently will attend a separate screening appointment. A full outline of the recruitment procedure is detailed in figure 1. All participants will be reimbursed for travel expenses.

Due to the paucity of research exploring the influence of medication on real-world mobility in PwP using a DHTS, there was insufficient data to inform a reliable power calculation before the study commenced. Therefore, a convenience sample size of 55 was defined according to COnsensus-based Standards for selection of health Measurement Instruments guidelines,[29] with the plan to assess this once 50% of data was collected. The sample size was evaluated using data collected in 30 participants, with GPower V.3.1, assuming an effect size of 0.44 for a change in mobility (gait speed, shown to be improved by levodopa in lab studies)[16] related to off versus on state (repeated paired t-test, alpha of 0.05 and beta (power) of 0.95). This highlighted that our sample size of 55 participants was appropriate.

### Participants and eligibility criteria

Participants with mild to moderate disease severity will be recruited (Hoehn and Yahr stages I–III). The eligibility criteria (inclusion and exclusion) are in agreement with the Mobilise-D study and are detailed in table 1. Potential participants will not be recruited if they have received and/or are due to receive a medication review within 4 weeks of the study assessment. Individuals with more advanced Parkinson's (Hoehn and Yahr stages IV and V) often possess a decreased ability to efficiently perform activities of daily living and experience cognitive impairment. These factors may induce difficulties to monitor mobility and alter the participant's ability to interact with the DHTS, respectively. Consequently, only individuals at Hoehn and Yahr stagess I–III will be recruited in this study.

### Study assessments

At the beginning of the study, all participants will complete a clinic-based assessment including a range of general, clinical, Parkinson's-specific, and physical assessments (box 1). These assessments will be performed during a Mobilise-D visit or within 14 days of the screening appointment for those recruited independently. Following assessment completion, participants will be equipped with a novel DHTS to wear and interact with over seven consecutive days and will complete a motor complications diary (ON-OFF-Dyskinesia) each day. Participants will be free to withdraw from the study at any time, without providing reasoning and they will be given a contact point to obtain further study information.

Researchers who collect data pertaining to outcome measures will receive appropriate training before data collection (eg, Good Clinical Practice training, informed consent in clinical research training and Movement Disorder Society-Unified Parkinson's Disease Rating Scale and Montreal Cognitive Assessment training). To promote participant recruitment and retention, participants will be given detailed oral and written instructions via a PIS. The PIS will include details for day-to-day use of the devices (ie, how to charge the devices and how to respond to medication notifications) and contact details for the research team should any issues or questions arise.

### Seven-day remote monitoring period

Primary outcome measures include self-reported medication adherence (ie, frequency, intake time and dosage of medication) captured by the smartwatch and walking speed, quantified via the IMU. Secondary outcome measures include macro outcomes (related to volume (eg, step count), pattern (eg, walking bout length) and variability (eg, walking bout length variability) of walking activity and micro outcomes (related to pace (eg, gait speed), rhythm (eg, step time), variability (eg, step time variability), asymmetry (eg, step time asymmetry) and postural control (eg, step length asymmetry)).[30–33] DMOs obtained during the 7-day monitoring period. In addition, contextual factors captured on the smartphone,

including geolocation (probability of being indoors/outdoors), number of steps outside the home, stay points, common paths and weather conditions (for more detail, see Smartphone section below).

### Measurement tools and data collection

To collect data on self-reported medication adherence, DMOs and contextual insights, participants will interact with the novel DHTS (smartphone, smartwatch and IMU) over seven consecutive days. This monitoring period will commence following a Mobilise-D appointment or following the initial assessment for those recruited independently.

### Smartphone to collect contextual data

As this study will be carried out remotely, in the participant's home and their community, contextual factors such as weather conditions and being indoors or outdoors, may influence DMOs. Therefore, capturing these contextual factors will help better describe, understand and control for these aspects when looking at the effect of medication on DMOs. To quantify the effect of these confounding contextual factors on DMOs, participants will be advised to carry a smartphone (Samsung Galaxy S9, S10 or S21) whenever outside their house, making sure it is switched on at all times and fully charged. Participants will be provided with a charger and will be advised to charge the smartphone every night throughout the monitoring period.

The custom-made Aeqora application extension will be preinstalled onto each smartphone (Department of Computer Science, The University of Sheffield, UK). This application will use the phone's sensors to capture contextual information, such as, geolocation, stay points, common paths, number of steps and weather conditions. The application has been developed as an Android mobile application. It is made up of three components: (1) the core tracker, (2) the interface and (3) the server infrastructure, which collects data across users.[34] The core tracker has been adapted from a library developed by the University of Sheffield.[35] This tracker identifies the type of activity (eg, walking) from the phone's internal sensors (accelerometer and gyroscope). It continually operates in the background to sense mobility features using these sensors and location services, such as Bluetooth.[34] To identify the weather conditions at the time of measurement, the closest weather station to the participant's location will be used. Raw sensor data will be collected and stored in real-time onto a local database which is synchronised with a remote database when the phone is recharging. No participant identifiable information will be sent to the server. The data are synched to the remote database via a Secure Socket Layer (SSL) and Transports Layer Security V.3.0 protocol. To ensure privacy and security when communicating sensitive data an SSL/TSL certificate is exchanged between client and server, which establishes the identity and trust between them.

### Smartwatch to assess real-world, self-reported medication adherence

To assess real-world, self-reported medication adherence, participants will wear and interact with a smartwatch (TicWatch Pro V.3, Mobvoi) throughout the monitoring period. During the clinic visit, a researcher will manually input the participant's medication reminders onto the Aeqora application. This application will then send notifications from the smartphone to the smartwatch to remind participants of their Parkinson's-specific medication intake times and allow them to log the times at which their medication was taken, clicking either 'yes' or 'no' on the watch face when prompted (figure 2). The watch will also allow participants to log any additional Parkinson's-specific medication they have taken ('as required'). Participants will be provided with a smartwatch charger and will be required to remove and charge the smartwatch overnight throughout the monitoring period. Common, validated methods for monitoring medication adherence rely on questionnaires and diary completion. These place extra burden onto participants and are often subjective. For this reason, this study uses the objective method of responding to notifications on a smartwatch, as this provides objective, real-time insights into medication adherence and minimises participant burden.

### IMU to quantify DMOs

To quantify primary and secondary DMOs, participants will continuously wear an IMU (Axivity AX6, Axivity, Newcastle upon Tyne, UK, including triaxial accelerometer and gyroscope, dimensions 23×32.5×8.9 mm, mass 11 g, frequency 100 Hz, gyroscope range±2000°/s and accelerometer range ±8 g) throughout the monitoring period. The IMU will be secured to the participant's lower back, at the fifth lumbar vertebrae.[34] It will be affixed using a hydrogel/hydrocolloid adhesive and a hypoallergenic plaster patch. Participants will be advised to continue with their usual activities and not change their routine. Throughout the monitoring period, all IMU data will be stored locally on the IMU hardware. After the monitoring period, participants will return the IMUs via prepaid envelopes and data will be downloaded and uploaded onto the eScience (e-SC) (a cloud-based storage platform) CiC Data Management platform[36 37] (see data management section). The IMU will be worn at all times, including during the night and although the IMU is waterproof participants will be advised to remove it during prolonged water exposure (eg, swimming, taking a bath).

### Diary to monitor motor complications

During the 7-day monitoring period, participants will fill in a motor complications diary (paper based) for each day, indicating their 'OFF-status' (when they feel their medication is not working) and whether they experienced dyskinesia (involuntary movements). The diary will record data over 16 hours each day from 6:00 to 22:00 hours, from days 1 to 7 (online supplemental material 4).

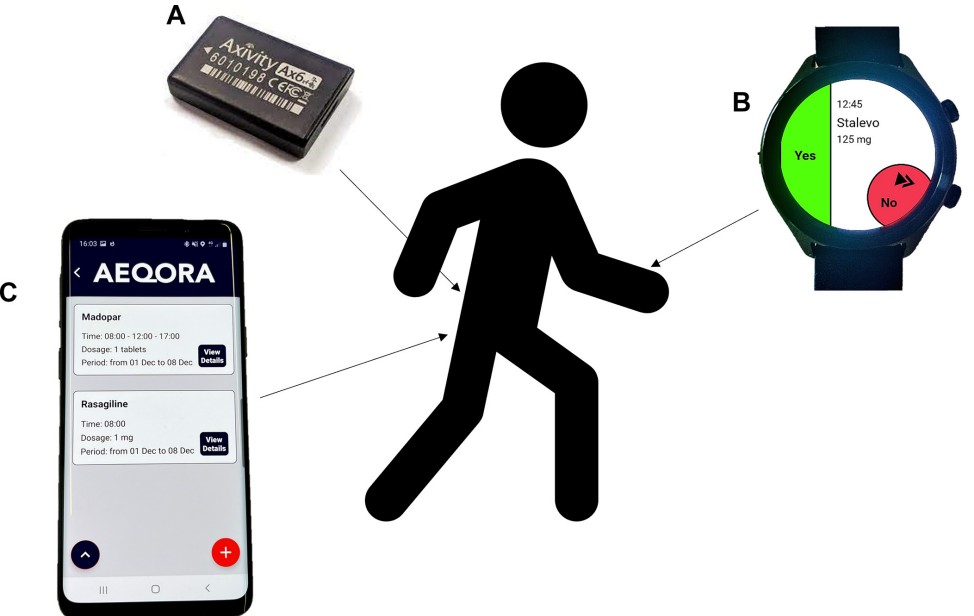

**Figure 2** Digital Health Technology System. (A) Inertial measurement unit, which monitors mobility. (B) Smartwatch which assesses self-reported medication adherence, by notifying participants when to take their medication and allowing them to log the time at which they took their medication. (C) Smartphone which collects contextual data.

## Assessment of participants' experience with the DHTS
### Questionnaire to evaluate usability of the DHTS
To evaluate the participants' experience with the DHTS, participants will complete an adapted version of Rabinovich et al's 12-item usability questionnaire at the end of the monitoring period (online supplemental material 5).[38] This questionnaire investigates usability on a 5-point ordinal scale, from 5 (most favourable) to 1 (least favourable), and 'no opinion' scored as 3. The questions identify participant's attitude towards the DHTS, its ease of use and comfort, such as, 'I experienced technical problems' and 'The monitors interfered with my normal activity'. Participants will also rate the DHTS on a scale from 0 (worst score) to 100 (best score).

Qualitative feedback on the DHTS and its individual devices will be collected via open-text questions added to the end of the questionnaire (online supplemental material 5). These questions require participants to 'give any other comments on the DHTS and its components', to describe the 'problems' they experienced when using the DHTS and its components, as well as any features they liked about the system and its individual devices. All devices will be returned at the end of the assessment period, in addition to the questionnaire and motor complications diary, using prepaid tracked envelopes.

### Feasibility of DHTS and motor complications diary
To assess whether the DHTS and motor complications diary are feasible for everyday use in PwP, this study will assess their feasibility (see data analysis section for details).

## Patient and public involvement
As this study is a substudy of the 'Mobilise-D—Clinical Validation Study' insights from patient and public advisory groups which informed the Mobilise-D study were used to inform the development of this study. The Mobilise-D Study PIS was reviewed and modified by relevant patient groups and was used as a guide for this study's PIS. Participants in this study will provide feedback on usability of the DHTS, which will inform future studies. Study insights will be fed back to patient stakeholder groups within the Mobilise-D consortium.

## Data and statistical analysis
### Data analysis
To quantify DMOs, mobility data from the IMUs will be analysed using validated Matlab algorithms (R2019a, Mathworks, California, USA).[37 39] To investigate the effect of medication on DMOs, medication intake timestamps (captured via the smartwatch) and mobility data from the IMU will be used. The effect of contextual factors (eg, weather conditions and probability of being indoors/outdoors) on DMOs will be considered when interpreting the medication and mobility data.

Responses to open-text questions will be analysed using a hybrid approach, with both deductive and inductive methods used to analyse responses. Responses will be grouped into three themes: feasibility, usability and recommendations for improvement. Once grouped, key subthemes within each group (ie, ease of use for the usability theme) will be identified following the same approach.

To assess the feasibility of the DHTS devices and the motor complications diary, we will assess whether the intended data was collected; specifically contextual data, self-reported medication adherence data, and mobility data for the smartphone, smartwatch, and IMU, respectively; and motor complications data for the motor complications diary.[40] A summary of feasibility measures and their outcomes is included in table 2.

**Table 2** Summary of feasibility and outcomes extracted from DHTS and motor complications diary

| Device | Measure of feasibility | Outcomes extracted |
|---|---|---|
| Smartphone | Percentage of datasets collected over 7 days and percentage of days missing | Number of steps taken outside the participant's home, per day |
| Smartwatch | Percentage of participants interacting with the smartwatch over 7-day monitoring period | No of interactions recorded on smartwatch |
| Inertial Measurement Unit | Percentage of datasets collected over 7-day monitoring period | Walking speed and number of steps per day |
| Motor complications diary | Percentage of diaries returned and legible | Time spent in an ON/OFF state or experiencing dyskinesia |

DHTS, digital health technology system; IMU, inertial measurement unit.

Only participants whose mobility data are collected for at least 3 days will be included in analysis. If participants do not complete one of the study assessments (eg, Short Physical Performance Battery) or their data from the devices is not collected, their remaining data will still be included in analysis. Assuming missing data is at random, modern statistical techniques will be used to handle missing data and will be compared with a complete case approach.[41]

### Statistical analysis

Participants' descriptive, clinical and Parkinson's specific characteristics, as well as data derived from the DHTS and motor complications diary data, will be analysed using descriptive statistics, including percentage and frequency, mean and SD or median and IQR, as appropriate.

Functional data analysis and deep learning techniques[42] will be used to model and predict DMO response to medication, and potential effects of contextual factors. DMO trends will be analysed between medication intake times and wearing-off effects will be evaluated via a customised scoring system. This scoring system will identify the wear off periods, grouping them relative to the strength of their effect. Non-linear time series analysis will assess changes in DMOs between medication intakes.[39 43] Data collected using motor complications diaries will be used to optimise time series modelling of DMOs.

Should the data collected require an adjusted methodology, proposed statistical methods will be amended following completion of data collection.

### Data management

All aspects of data management will adhere to the FAIR guiding principles, ensuring that data are findable, accessible, interoperable and reusable,[44] as well as being attributable, legible, contemporaneous, original and accurate.[45] The CiC data management platform has been created on the open source, cloud-based platform e-Science Central (e-SC).[36 37] This platform securely stores and shares data and has data processing capabilities.

This is a low risk, observational study which does not require a data monitoring committee. The data management group (SMG) will be responsible for managing data sharing requests, and on data requests will meet with external researchers. This protocol received input from Parkinson's and clinical experts (LR and AJY); technical experts (SDD, LA and JQS) gave their input on data collection methods, outcomes and eligibility criteria.

### Data collection, storage and archiving

To maintain confidentiality, participant data will be collected in a coded, anonymised manner using unique identification codes. Data will be directly uploaded onto the e-SC CiC platform or via third-party platforms (eg, weather data). All researchers will be trained to correctly enter and upload data to relevant platforms. All data collected on paper case report forms will be signed, dated, scanned and uploaded onto a secure network drive with restricted access. The e-SC CiC platform will be implemented using secure Amazon Web Services located within the European Union. The clinical site will keep original records of all paper forms in secure conditions until the study is completed. Long-term data storage will be in accordance with the Newcastle Joint Office policy for archiving, with data security arrangements made in accordance with Newcastle upon Tyne National Health Service (NHS) Foundation Trust policies. The clinical site will be responsible for archiving study documents for a period which is in keeping with the institutional or national guidelines pertaining to the site. Detailed information on the data transfer pipeline is summarised in online supplemental material 6.

### ETHICS AND DISSEMINATION

This single-centre study is sponsored by the Newcastle upon Tyne Hospitals NHS Foundation Trust (NuTH) UK. Ethical approval was obtained from the London—142 Westminster Research Ethics Committee (REC reference: 21/PR/0469). The study will be conducted in accordance with Good Clinical Practice standards, the Declaration of Helsinki and The European Code of Conduct for Research Integrity.[46] This study will be included in the NIHR Clinical Network Portfolio and has been registered in the ISRCTN registry (ISRCTN number: 13156149). Informed consent will be obtained by a trained researcher. The NHS indemnity scheme and Newcastle University are liable for harm to participants arising from the management and conduct of the study, and from its design, respectively.

As this is an observational study, there is no adverse event reporting planned. Should any serious adverse events (SAEs) which are related to the administration of the research procedure occur, the sponsor and chief investigator will be informed. On behalf of the sponsor, the Newcastle Joint Research Office Governance Team

will be responsible for assessing protocol deviations, serious breaches and violations, as well as reporting Serious Breaches to the REC. The chief investigator is responsible for reporting protocol deviations and violations. Any protocol amendments will be communicated to the sponsor with all required amendment documents completed. The sponsor will be required to agree and accept any modifications, before amendments can be submitted to the NHS REC for their approval.

## Dissemination policy

Newsletters will be sent to participants to inform them of study outcomes. Research will be disseminated through peer-reviewed scientific journals, internal reports, conference presentations and relevant websites. We will follow the International Committee of Medical Journal Editors recommendations for authorship eligibility.

## Access to data

We plan to indefinitely maintain the complete, anonymised dataset on the e-SC platform. On study completion, this dataset will be made available to the wider research community; on request and will be shared with approved collaborators nationally and internationally for peer review, scientifically sound studies. Participants will have the right to access a copy of their data or request its removal following an appropriate request. These rights will terminate following dataset anonymisation and after the study key code is destroyed. The lead clinical investigator has overall responsibility for study execution and can access the full dataset. Data analysts, who are responsible for implementation of data protocol, will have access to the full-anonymised dataset. Statisticians will only have access to the full-anonymised datasets and will not have access to raw data files. Access to study data will be given to authorised representatives from the sponsor, host institution and regulatory authorities to allow study related audits, monitoring and inspections—in line with participant consent.

## Study status

The first phase of recruitment has finished, with data collected in 30 PwP. Recruitment for the second phase of the study will start in February 2023 and will continue until June 2024 or until the desired sample size has been reached.

**Author affiliations**
[1]Translational and Clinical Research Institute, Faculty of Medical Sciences, Newcastle University, Newcastle upon Tyne, UK
[2]Dipartimento di Informatica, Università di Torino, Torino, Italy
[3]Department of Computer Science and INSIGNEO Institute for in silico Medicine, The University of Sheffield, Sheffield, UK
[4]McRoberts BV, The Haag, The Netherlands
[5]Department of Statistics and Data Science, Southern University of Science and Technology, Shenzhen, Guangdong, China
[6]National Center for Applied Mathematics, Shenzhen, Guangdong, China
[7]Newcastle Upon Tyne Hospitals NHS Foundation Trust, Newcastle Upon Tyne, UK
[8]Based at The Newcastle upon Tyne Hospitals NHS Foundation Trust, NIHR Newcastle Biomedical Research Centre, Newcastle University, Newcastle upon Tyne, UK

**Acknowledgements** The authors would like to acknowledge and thank all collaborators of this study. We gratefully acknowledge the contribution of our much-missed colleague Dr Richard Dodds, who sadly passed away before this study could be completed.

**Contributors** Study design: SDD, LA, AJY and LR. Ethical approval and clinical oversight: SDD, LA, AJY, LR. Manuscript initial drafting: EP and HD. Figures and tables preparation: EP and HD. Manuscript revision: EP, HD, SDD and LA. Intellectual contribution: EP, HD, HGBB, FC, NI, JE, MN, JQS, AJY, LR, LA and SDD. All authors have provided critical intellectual input during manuscript revision. All authors have reviewed and approved the submitted manuscript. The corresponding author attests that all listed authors meet authorship criteria and that no others meeting the criteria have been omitted.

**Funding** The study is cofunded by the Medical Research Council (MRC) Newcastle University (MC_PC_19047, NU-002884), the UK Research Innovation, Engineering and Physical Sciences Research Council (UKRI EPSRC) and the National Institute of Health Research (NIHR) (Grant Ref: EP/W031590/1, 'Transforming care and health at home and enabling independence'). This study was also supported by the Mobilise-D project which has received funding from the Innovative Medicines Initiative 2 Joint Undertaking, under the grant agreement No. 820820. This JU receives support from the European Union's Horizon 2020 research and innovation program and the European Federation of Pharmaceutical Industries and Associations (EFPIA). The research was supported by the National Institute for Health and Care Research (NIHR) Newcastle Biomedical Research Centre based at The Newcastle upon Tyne Hospitals NHS Foundation Trust, Newcastle University and the Cumbria, Northumberland and Tyne and Wear (CNTW) NHS Foundation Trust. The research was also supported by NIHR Newcastle Clinical Research Facility (CRF) Infrastructure funding. All opinions are those of the authors, not the funders.

**Disclaimer** Views expressed are those of the authors and do not necessarily reflect those of the NHS, the NIHR, UKRI EPSRC, Department of Health, the IMI, the European Union, the EFPIA or Associated Partners.

**Competing interests** FC was CEO and shareholder of Aeqora. JE and MN are employees of McRoberts. SDD reports consultancy activity with Hoffmann-La Roche outside of this study.

**Patient and public involvement** Patients and/or the public were involved in the design, or conduct, or reporting, or dissemination plans of this research. Refer to the Methods section for further details.

**Patient consent for publication** Not applicable.

**Provenance and peer review** Not commissioned; externally peer reviewed.

**ORCID iDs**
Emma Packer http://orcid.org/0000-0002-0491-3726
Martijn Niessen http://orcid.org/0000-0003-3515-4731

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
