## [Reviewer comments · BMJ Open]

ARTICLE DETAILS

TITLE (PROVISIONAL)	Translating digital healthcare to enhance clinical management: a protocol for an observational study utilising a digital health technology system to monitor medication adherence and its effect on mobility in people with Parkinson's
AUTHORS	Packer, Emma; Debelle, Héloïse; Bailey, Harry; Ciravegna, Fabio; Ireson, Neil; Evers, Jordi; Niessen, Martijn; Shi, Jian Qing; Yarnall, Alison J; Rochester, Lynn; Alcock, Lisa; Del Din, Silvia

VERSION 1 – REVIEW

REVIEWER	Olmo, Gabriella Politecnico di Torino
REVIEW RETURNED	27-Apr-2023

GENERAL COMMENTS	This cross-sectional observational study protocol has the objective of evaluating the adherence to medical therapy and the effects of medication on mobility in people with Parkinson's disease (PD). Data are collected during activities of daily living for seven days. The medication intakes are recorded using a smartwatch, which also sends reminders and acknowledgments). An IMU is used to quantify the so-called "digital movement outcomes" (DMOs: e.g., walking speed and other movement measures). Moreover, "contextual factors" such as geolocation (probability of being indoors/outdoors), number of steps outside home, common paths, weather conditions are captured by the smartphone sensors via a proper Android application. During the seven-day monitoring period, participants are asked to fill in a motor complications diary indicating ON/OFF periods and dyskinesia. Finally, the response of DMOs to medication, their relationship with motor complications and the dependency on contextual factors are assessed using functional data analysis, statistical instruments and deep learning techniques. At the end of the monitoring period, the acceptability and usability of the system will be assessed by means of a questionnaire. The study involves 55 PD patients, identified and recruited between June 2021 and June 2024. The sample size of 55 was defined according to proper guidelines. The eligibility criteria and the enrolling procedure are clearly stated. Information about the required consent and data treatment are reported. In my opinion, the study protocol is sound; the research objectives are clearly stated and scientifically sound.
---

	I have some comments that may be considered by the Authors to improve the readability of the manuscript. [ ] I understand that the enrolling period is 3 years and almost 2/3 already elapsed. It seems quite a long period to recruit 55 patients. Moreover, I don't see a clear timetable of the study. When is it assumed to be concluded? [ ] Another point is that the use of both a smartphone and an IMU seems a bit redundant. As the inertial sensors of the smartphone are employed, maybe the same data could have been measured using the IMU itself. Please clarify the usefulness of having one more instrument, and the expected impact of the so-called "contextual factors".
--	--

REVIEWER	Pérez-Lloret, Santiago University of Buenos Aires
REVIEW RETURNED	06-Jun-2023

GENERAL COMMENTS	This study aims at assessing an integrative Digital Health Technology platform in patients with Parkinson's Disease (PD). The extent of the assessment at the clinical setting of patients with PD is forcefully limited. Therefore, we need better tools to assess patients at their homes. The protocol is overall ok. However, I think that the sample size is insufficient to obtain any meaningful clinical conclusions. I understand that the study aims at assessing the feasibility of the DHT platform, but this limitation should be acknowledged. I also suggest including a validated measure of medication adherence to contrast with the digital assessment.
--

VERSION 1 – AUTHOR RESPONSE

Reviewer 1	
Comments	Reply
I understand that the enrolling period is 3 years and almost 2/3 already elapsed. It seems quite a long period to recruit 55 patients. Moreover, I don't see a clear timetable of the study. When is it assumed to be concluded?	Thank you for your comment. The study is due to conclude in June 2024. We appreciate that three years is a long period to recruit 55 participants, however, recruitment commenced in the height of the pandemic, when individuals were less mobile and fewer people took part in research, due to government lockdowns and guidelines to reduce outdoor activity. As our research is specifically exploring mobility, we wanted to ensure we could capture enough data on this, which may not have been the case if all participants were studied during the pandemic. Additionally, as there was no indication of when lockdowns would cease and whether there would be additional lockdowns, we factored this into our recruitment period. Therefore, to ensure our sample was representative, and ensure that we could capture enough mobility data, we

	employed a longer recruitment period. We have since recruited 35 participants, with additional participants booked in. We have updated the methods section to include this explanation and ensure this is clear to readers: “As recruitment commenced in the height of the COVID-19 pandemic - i.e., when individuals were less mobile, less likely to partake in research, and there was insecurity surrounding future lockdowns - an extended recruitment period was afforded to this study. This was to ensure to our sample size was achieved and results were representative.”
Another point is that the use of both a smartphone and an IMU seems a bit redundant. As the inertial sensors of the smartphone are employed, maybe the same data could have been measured using the IMU itself. Please clarify the usefulness of having one more instrument, and the expected impact of the so-called “contextual factors”	The smartphone was employed to collect data on contextual factors such as the weather, probability of being indoors or outdoors, and to record medication adherence by sending and logging medication reminders to the smartwatch via a custom developed software Application, both factors which the IMU could not measure. Whereas, the IMU was employed to measure mobility using validated algorithms. Understanding the environment (e.g. indoor vs outdoor, weather) in which participants were in whilst their mobility is assessed, is an important confounding aspect to assess, as factors such as rain will cause individual’s to walk more slowly. Although the smartphone can collect mobility data such as number of steps, this information is very top level and does not provide us with the granular insight into digital mobility outcomes like step length and gait speed, which we require. This section now reads as follows: “As this study will be carried out remotely, in the participant’s home and their community, contextual factors such as weather conditions and being indoors or outdoors, may influence DMOs. Therefore, capturing these confounding factors will help better describe, understand and control for these aspects when looking at the effect of medication on DMOs.”
Reviewer 2	
Comments	Reply
I think that the sample size is insufficient to obtain any meaningful clinical conclusions. I understand that the study aims at assessing the	Thank you for your comment, due to the paucity of research which utilises digital health technology to explore how medication influences real-world mobility in people with

feasibility of the DHT platform, but this limitation should be acknowledged.	Parkinson's, a power calculation could not be performed, with the convenience sample size of 55 defined according to COSMIN guidelines. The sample size was evaluated with GPower v3.1, assuming an effect size of 0.44 for a change in mobility (gait speed, shown to be improved by levodopa in lab studies [ref - Smulders]) related to off vs on state (repeated paired t-test, alpha of 0.05 and beta (power) of 0.95). The text now reads as follows: "Due to the paucity of research exploring the influence of medication on real-world mobility in PwP using a DHTS, there was insufficient data to inform a reliable power calculation before the study commenced. Therefore, a convenience sample size of 55 was defined according to COnsensus-based Standards for selection of health Measurement Instruments (COSMIN) guidelines (29), with the plan to assess this once 50% of data was collected. The sample size was evaluated using data collected in 30 participants, with GPower v3.1, assuming an effect size of 0.44 for a change in mobility (gait speed, shown to be improved by levodopa in lab studies (16) related to off vs on state (repeated paired t-test, alpha of 0.05 and beta (power) of 0.95). This highlighted that our sample size of 55 participants was appropriate."
I also suggest including a validated measure of medication adherence to contrast with the digital assessment.	Current validated measures of medication adherence generally rely on self-reported questionnaires or diaries. However, these have poor recall bias and place greater burden on participants. They can also be inaccurate if participants forget to fill the diary out and fill it out at a later date. For this reason we chose an objective method to monitor medication adherence, where participants were notified on the smartwatch to take medication, clicking either yes, or no. Using the smartwatch, participants could also input if they had taken additional medications outside of their regimen. This meant we collected real-time, objective measures of medication intake and reduced participant burden. We did not use a validated measure of medication adherence, as we did not want to place greater burden on participants and other similar methods, such as Personal Kinetigraphs (PKG) have successfully collected

	medication adherence data in a very similar manner. We have added a section to the paper to address this: “Common, validated methods for monitoring medication adherence rely on questionnaires and diary completion. These place extra burden onto participants and are often subjective. For this reason, this study utilises the objective method of responding to notifications on a smartwatch, as this provides objective, real-time insights into medication adherence, and minimises participant-burden.”
--	--

VERSION 2 – REVIEW

REVIEWER	Olmo, Gabriella Politecnico di Torino
REVIEW RETURNED	24-Jul-2023

GENERAL COMMENTS	The Authors have properly taken my issues into account. I have no further comment.
--

REVIEWER	Pérez-Lloret, Santiago University of Buenos Aires
REVIEW RETURNED	10-Jul-2023

GENERAL COMMENTS	I have no further comments.
-----------------------------